# Machinery Prognostics and High-Dimensional Data Feature Extraction Based on a Transformer Self-Attention Transfer Network

**DOI:** 10.3390/s23229190

**Published:** 2023-11-15

**Authors:** Shilong Sun, Tengyi Peng, Haodong Huang

**Affiliations:** 1Guangdong Key Laboratory of Intelligent Morphing Mechanisms and Adaptive Robotics, Shenzhen 518055, China; 20s053007@stu.hit.edu.cn (T.P.); 21s053030@stu.hit.edu.cn (H.H.); 2School of Mechanical Engineering and Automation, Harbin Institute of Technology, Shenzhen 518055, China

**Keywords:** feature extraction, prognostics, self-attention transfer network, high-dimensional data, remaining useful life prediction

## Abstract

Machinery degradation assessment can offer meaningful prognosis and health management information. Although numerous machine prediction models based on artificial intelligence have emerged in recent years, they still face a series of challenges: (1) Many models continue to rely on manual feature extraction. (2) Deep learning models still struggle with long sequence prediction tasks. (3) Health indicators are inefficient for remaining useful life (RUL) prediction with cross-operational environments when dealing with high-dimensional datasets as inputs. This research proposes a health indicator construction methodology based on a transformer self-attention transfer network (TSTN). This methodology can directly deal with the high-dimensional raw dataset and keep all the information without missing when the signals are taken as the input of the diagnosis and prognosis model. First, we design an encoder with a long-term and short-term self-attention mechanism to capture crucial time-varying information from a high-dimensional dataset. Second, we propose an estimator that can map the embedding from the encoder output to the estimated degradation trends. Then, we present a domain discriminator to extract invariant features from different machine operating conditions. Case studies were carried out using the FEMTO-ST bearing dataset, and the Monte Carlo method was employed for RUL prediction during the degradation process. When compared to other established techniques such as the RNN-based RUL prediction method, convolutional LSTM network, Bi-directional LSTM network with attention mechanism, and the traditional RUL prediction method based on vibration frequency anomaly detection and survival time ratio, our proposed TSTN method demonstrates superior RUL prediction accuracy with a notable SCORE of 0.4017. These results underscore the significant advantages and potential of the TSTN approach over other state-of-the-art techniques.

## 1. Introduction

Machine condition prognostics is the critical part of an intelligent health management (PHM) system, which aims to predict a machine’s remaining useful life (RUL) based on condition monitoring information [1]. The general PHM procedures include the construction of health indicators (HIs) and RUL prediction. The HI is a crucial variable that indicates the current machine health condition, and also it represents the information extracted from sensor data and provides degradation trends for RUL prediction.

The HI construction process is called data fusion and has three categories: feature-level, decision-level, and data-level fusion [2]. Feature-level fusion methods rely on prior knowledge of degradation mechanisms and physical models. Ma [3] reported a multiple-view feature fusion method for predicting the RUL of lithium-ion batteries (LiBs). Decision-level techniques fuse high-level decisions based on individual sensor data and do not depend on raw-signal feature extraction. Lupea [4] developed a system utilizing features from vibration signals to detect mounting defects on a rotating test rig, with the quadratic SVM classifier emerging as the top performer Wei [5] proposed a decision-level data fusion method to map a unique sensor signal onto reliable data to improve the capability of the quality control system in additive manufacturing and RUL estimation for aircraft engines. Data-level fusion methods find the embedding feature suitable for a task from raw data. They can monitor the machine system state based on the requirements of an effective aero-engine prognostic and also the monitoring task has strong versatility. Chen [6] proposed an improved HI fusion method for generating a degradation tendency tracking strategy to predict the gear’s RUL. Wang [7] extended the extreme learning machine to an interpretable neural network structure, which can automatically localize informative frequency bands and construct HI for machine condition monitoring. RUL prediction reveals the remaining operating time before equipment requires maintenance. They can be classified into four categories: physics model-based, statistical model-based, artificial intelligence-based, and hybrid methods [8]. Many recent studies have focused on artificial intelligence-based machine RUL prediction methods such as convolutional neural networks (CNNs) [9], long short-term memory (LSTM) recurrent networks [10], and gated recurrent (GRU) networks [11]. Recurrent neural networks (RNNs) have gradually become the most popular of these methods. Many scholars have focused on LSTM recurrent networks and GRU networks to address the vanishing gradient problem. Xiang [12] added an attention mechanism to the basis of an ordered, updated LSTM network, which further improved the robustness and accuracy of the LSTM network-based RUL prediction model.

Although these methods can achieve an effective machine prognostic, most artificial intelligent-based models rely on manual feature extraction (HI construction). Manual feature extraction inevitably leads to information loss, which has a negative influence on prognostics. Several studies have focused on allowing neural networks to extract features automatically from the original input, a procedure that can avoid input information loss from manual feature extraction. In the fault diagnosis field, artificial intelligence-based models exhibit excellent fault diagnosis performance with the original vibration signal input [13]. Ambrożkiewicz [14] presented an intelligent approach to detect the radial internal clearance values of rolling bearings by analyzing short-time intervals and calculating selected indicators, later enhancing classification accuracy using Variational Mode Decomposition (VMD). They can directly extract disguisable fault features from unlabeled vibration signals [15]. These methods mainly utilize CNNs to realize automatic feature extraction. Therefore, several researchers have attempted to utilize CNNs to extract degradation features for predictive purposes. Xu [16] applied a dilated CNN to the field of prognostics, used five convolutional layers to extract features from the original signal, and combined them with a fully connected network to realize effective prognostics. Li [17] proposed a multivariable machine predictive method based on a deep convolutional network. The proposed method uses the time-window method to construct 2D data as convolutional network input. Ren [18] built a spectrum principal energy vector from a raw vibration signal as a CNN input for bearing prognostics. CNNs demonstrate a strong capability in high-dimensional input situations but are not good at dealing with long-term series prognostics tasks. RNNs can easily construct long-term relationships but cannot directly utilize the abundant long-term information owing to their limited in-network processing capacity. Thus, this study proposes building a network that can directly deal with high-dimensional, long-term, time-series data for machine prognostics. The aim was to establish the long-term degradation relationship for prognostics from a large amount of raw data without relying on manual feature extraction and HI construction.

Another non-negligible defect of the existing prognostics methods is that all degradation datasets satisfy independent and identically distributed conditions. Due to the operating condition and fault type variation, a distribution discrepancy generally exists between degradation datasets (each degradation dataset is an independent domain), leading to performance fluctuation in prognostics methods. Hadi [19] introduced two automated machine-learning models aimed at precisely identifying various ball-bearing faults. Using the CWRU bearing faults dataset for evaluation, their study emphasized the potential of AutoML techniques in IIoT applications, especially valuable for industries where unscheduled downtimes can be costly. Transfer learning (TL) is introduced to help artificial intelligence-based prognostics methods extract domain-varied features and achieve effective outcomes under cross-operating conditions. TL can utilize the knowledge learned in previous tasks for new tasks by removing the domain invariance feature [20], which is widely used in fault-diagnosis tasks. In recent years, many researchers have focused on TL application in the prognostics field to achieve effective cross-operating condition prognostics. For example, Wen [21] utilized a domain adversarial neural network structure to solve the crossing domain prognostic problem. Roberto [22] proposed a domain adversarial LSTM neural network that achieved an effective aero-engine prognosis. Mao [23] performed a transfer component analysis that sequentially adjusts the features of current testing bearings from auxiliary bearings to enhance prognostics accuracy and numerical stability. This study introduces TL to extract the general representation of bearing degradation data from different operating conditions and the final fault types to achieve prognostics in cross-operating conditions. Figure 1 shows a general transfer learning algorithm for the cross-operating conditions’ HIs.

Transformer [24] is a popular multi-modal universal architecture neural network architecture. The transformer utilizes a self-attention mechanism to capture the long-term dependence (spatial dependence) information between input elements in a sequence. It uses the full sequence input for each inference; therefore, it is less affected by the sequence length than traditional methods (RNN and LSTM). This feature of the transformer network is suitable for the prognostic task. Zhang [25] proposed a dual-aspect transformer network to fuse the time steps and sensor information for long-time machine prognostic. Su [26] proposed a bearing prognostic method consisting of a transformer and LSTM, achieving effective RUL prediction. Thanks to the advantages of the transformer architecture in processing long series and high-dimensional features, it has the potential to become a well-data-driven prognostic tool. Therefore, the cross-domain prognostic based on a transformer architecture is studied.

To address the limitations introduced by the above issues concerning feature extraction, cross-operating conditions, and different data distributions, this study takes the FEMTO-ST bearing dataset as an example to explore the degradation process based on a transformer-based self-attention transfer learning network (TSTN). The method can automatically construct an HI from high-dimensional feature inputs and realize long-term information association to monitor machine conditions. The innovations and contributions of this study are summarized as follows:(1)Development of TSTN for Machine Prognostics:

We have introduced the Transformer-Based Self-Attention Transfer Learning Network (TSTN) as a dedicated solution for machine prognostics. TSTN leverages long-term, high-dimensional spectrum vectors as its input and directly produces a linear Health Index (HI) output, a numerical value ranging from 0 to 1. This HI value is straightforwardly compared to a failure threshold of 1. The core transformer architecture within TSTN plays a pivotal role in extracting critical features from extended time sequences.

(2)Incorporation of Long-term and Short-term Self-Attention Mechanisms:

TSTN incorporates both long-term and short-term self-attention mechanisms, empowering it to discern short-term and long-term fluctuations in machine conditions. By analyzing historical high-dimensional feature data in conjunction with current information, TSTN excels at identifying evolving machine states.

(3)Integration of Domain Adversarial Network (DAN) in TSTN:

To enhance TSTN’s robustness and versatility, we have integrated a Domain Adversarial Network (DAN) within its architecture. DAN effectively minimizes data disparities across various operational conditions, thus enabling TSTN to monitor machine states consistently across different scenarios and environments. This integration significantly extends TSTN’s applicability for cross-operation machine state monitoring.

The remainder of this paper is organized as follows. Section 2 introduces the preliminaries of the proposed method. The principle of the proposed algorithm is presented in Section 3. Section 4 describes the proposed model’s experimental study, and Section 5 summarizes this work.

## 2. The Related Work

This section reviews the basic architecture of the transformer network structure and adversarial domain structure.

### 2.1. Transformer Network Structure

Vaswani proposed a transformer network structure [24]. This network is used to solve the shortcomings of the sequential computation network; that is, the number of operations required to relate signals from two arbitrary input positions increases with the distance between positions. The critical part of the transformer is the self-attention layer, which consists of two sub-parts: the multi-head attention layer and the feedforward network (FFN). The structure of the self-attention layer is illustrated in Figure 2.

The critical operation of the self-attention layer is scaled dot-product attention (right side of Figure 2).

Assuming that the input data X consists of n patches, the i-th patch is denoted as xi, and the corresponding “query” (q∈ℝ1×dmodel), “keys” (k∈ℝ1×dmodel), and “values” (v∈ℝ1×dmodel) can be calculated through linear mapping (qi=WQ×xiT, ki=WK×xiT, vi=WV×xiT). 

In addition, WQ∈ℝdmodel×dpatch, WK∈ℝdmodel×dpatch, and WV∈ℝdmodel×dpatch were trainable variables. 

To improve the learning capability of the self-attention layer, k, v, and q are linearly projected h times, which is called the multi-head attention layer. For example, qi is decomposed into [qi,1,qi,2,⋯,qi,h], and the operations of ki and vi are similar to those of qi. j-th sub-parts of qi, ki, and vi are denoted as qi,j, ki,j, and vi,j, respectively. The scaled dot-product attention operation is
(1)Headi,j=att(qi,j,kdot,j,vi,j)≜softmax(qi,jkdot,jTdk)vi,j,
where kdot,j refers to all ki,j that must be calculated via the scaled dot-product attention operation. After the scaled dot-product attention operation, the output results of the multi-head attention layer are
(2)MultiHeadi=Concat(Head1,Head2,⋯,Headh)WO,
where WO∈ℝdpatch×dmodel represents the learnable linear projection. To facilitate expression, the operations (1) and (2) are summarized into one operation symbol SM(qi,j,kdot,j,vi,j). FFN consists of one hidden layer, and the density of the hidden layer is denoted as ddiff; the density of the output layer is dmodel.

### 2.2. Domain Adversarial Network

An adversarial domain network (DAN) is an effective TL method that can extract domain-invariant features [27], and its architecture is shown in Figure 3. The DAN introduces adversarial learning to achieve domain adaptation. In addition to the standard feed-forward feature extractor and label predictor, the DAN contains a domain classifier that connects to the feature extractor via a gradient reversal layer. During backpropagation-based training, the gradient reversal layer multiplies the gradient by a certain negative constant. The training process must minimize label prediction and domain classification losses. The feature distributions of all domains were similar to those of the domain classifier and the gradient reversal layer.

## 3. The Proposed TSTN

### 3.1. TSTN Structure

The proposed network structure for machine RUL prediction based on the transformer and multiple-source domain adaptation is shown in Figure 4. The proposed network consists of three subparts: an encoder, HI estimator, and a domain discriminator.

The input data of this network is xt. When data xt∈ℝ(m×n)×p are fed into the network p, a learnable patch x0 is added in front of vector xt and multiplies this vector p. The input sequence is X∈ℝ(1+m×n)×p. The learnable patch on the encoder output serves as the HI representation, connecting the HI estimator and domain discriminator. Learnable patches calculate self-attention with others to capture the long-term collected signal sequence’s high-dimensional feature (spectrum) change. The encoder of the proposed TSTN consists of a local, long-term, and short-term self-attention layer and a feed-forward network. For the ease of expression, Hinput∈ℝ(1+m×n)×p and Houtput∈ℝ(1+m×n)×p are denoted as encoder input and output, respectively.

It is well known that the datasets collected from different operating conditions and fault types are challenging in terms of satisfying the independent identically distribution (IID) property. Hence, this proposed method introduces a domain discriminator with a gradient reversal layer to make the HI representation distribution of different degradation datasets similar. This method can realize prognostics under cross-operating conditions. The encoder, HI estimator, and domain discriminator are introduced as follows. The detailed network settings are listed in Figure 4. In the training process, the forward data flow is plotted using blue arrows, and the backward gradient flow is plotted using orange arrows. The functions $L_{HI}$ and $L_{d}$ were added directly as $L=L_{HI}+L_{d}$ in the training process. Figure 4 displays the parameter setting of the proposed TSTN methodology.

**Query–key–value computation.** The encoder input $H_{\mathrm{input}}^{}$ consists of 1+m×n patches. The l-th patch collected in the s-th frame is $H_{\mathrm{input}}^{}$ denoted as $h_{s,l}$, and the query, key, and value vectors are $t_{index}$ computed by $q_{s,l}=W_{Q}\times h_{s,l}{}^{T}$, $k_{s,l}=W_{K}\times h_{s,l}{}^{T}$, and $v_{s,l}=W_{V}\times h_{s,l}{}^{T}$, respectively. Following the extended derivation in [28], denoting the s-th frame corresponding time is $t_{index}$, and the rotary position embedding in the proposed method as follows:(3)$$q_{s,l}^{R}=\left(\begin{array}{c} q_{1}\\ q_{2}\\ q_{3}\\ q_{4}\\ \vdots \\ q_{d_{\mathrm{model}-1}}\\ q_{d_{\mathrm{model}}} \end{array}\right)⊗\left(\begin{array}{c} \cos t_{index}\theta_{1}\\ \cos t_{index}\theta_{1}\\ \cos l\theta_{1}\\ \cos l\theta_{1}\\ \vdots \\ \cos l\theta_{p/4}\\ \cos l\theta_{p/4} \end{array}\right)+\left(\begin{array}{c} q_{2}\\ q_{1}\\ q_{4}\\ q_{3}\\ \vdots \\ q_{d_{\mathrm{model}}}\\ q_{d_{\mathrm{model}-1}} \end{array}\right)⊗\left(\begin{array}{c} -\sin t_{index}\theta_{1}\\ \sin t_{index}\theta_{1}\\ -\sin l\theta_{1}\\ \sin l\theta_{1}\\ \vdots \\ -\sin l\theta_{p/4}\\ \sin l\theta_{p/2} \end{array}\right),$$


The predefined parameter is $\Theta =\left\{\theta_{j}=10000^{-4\left(j-1\right)/d_{\mathrm{model}}},j\in \left[1,2,\cdots ,d_{\mathrm{model}}/4\right]\right\}$, and the calculation operation of $k_{s,l}^{R}$ is similar to that in (3). Using this position embedding method, the signal collected time information $t_{index}$ and the spectrum location information l of patch $h_{s,l}$ can be recognized using self-attention. The first learnable patch $h_{0,0}$ needs the use of the same method to generate $q_{0,0}^{R}$, $k_{0,0}^{R}$, and $v_{0,0}^{}$. Since the time-embedding information offers the time auxiliary information, private over-fitting $t_{index}$ will time a random value governed by N(1,0.003).

**Long-term, local, and short-term self-attention.** The dimensions of the input data $x_{t}$ are enormous. The number of calculations is large when self-attention is calculated for each patch, thereby confusing the network. We propose three sub-self-attention parts to allow the network to capture the degradation trend from the high-dimensional spectrum: local, long-term, and short-term self-attention.

To trace the long-term trend of machine conditions, we compute it by comparing each patch with all patches at the same spectrum location.
(4)$$a_{s,l}^{R}{}^{\left(\mathrm{Long}-\mathrm{term}\right)}=SM(q_{s,l}^{R},\left[k_{0,0}^{R},{\left\{k_{i,l}^{R}\right\}}_{i=1,\cdots ,m}\right],v_{s,l}^{}).$$

To learn the spectrum information from each collected signal, local self-attention operation only computes patches with the others collected simultaneously. The local self-attention operation is
(5)$$a_{s,l}^{R}{}^{\left(\mathrm{Local}\right)}=SM(q_{s,l}^{R},\left[k_{0,0}^{R},{\left\{k_{s,i}^{R}\right\}}_{i=1,\cdots ,n}\right],v_{s,l}^{}).$$

The rapid, short-term changes in the machine conditions can be computed as follows:(6)$$a_{s,l}^{R}{}^{\left(\mathrm{Short}-\mathrm{term}\right)}=SM(q_{s,l}^{R},\left[k_{0,0}^{R},{\left\{k_{i,j}^{R}\right\}}_{i=1,\cdots ,s;j=1,\cdots ,n}\right],v_{s,l}^{}),$$
where s denotes the first s frame on which we wish to focus. After calculating all patches $H_{\mathrm{input}}^{}$ via a self-attention operation, the output is represented as A. 

**Residual connection and layer normalization.** After the self-attention computation, the output of the attention layer is calculated via the $B=LayerNorm\left(A+H_{\mathrm{input}}^{}\right)$ residual connection [29] and layer normalization [30].**FFN and layer normalization.** The final layer of the encoder is the FFN and layer normalization; that is, $H_{\mathrm{output}}=LayerNorm\left(B+FFN(B)\right)$.

The feed-forward layer consists of an MLP with one hidden layer. The density of the hidden layer is denoted by $d_{\mathrm{diff}}=8\;p$, and the density of the output layer is denoted by p. Notably, the activation function of the hidden layer is GeGLU [31], and the output layer has no activation function. GeGLU introduced gates to modulate the linear projection, which can control the information that is not conducive to HI estimation passed on to the encoder.

Subsequently, all operations in $H_{\mathrm{input}}^{}$ are encoder outputs. To combine the long-term, local, and short-term self-attention into one encoder, $B^{\left(\mathrm{Long}-\mathrm{term}\right)}$ is fed back to calculate the local self-attention instead of being passed to the FFN. Hence, the new $Q^{R},K^{R}$ and V are generated from $B^{\left(\mathrm{Long}-\mathrm{term}\right)}$ and fed into Equation (5) to calculate local self-attention. The operation of short-term self-attention was similar to that of local self-attention.

**HI estimator.** An MLP with one hidden layer was connected to the learnable patch of the encoder output, and the MLP output was the HI estimated result $e_{HI}$.

To indicate HI easily and intuitively, the training label is defined by the index results from the normalized operating time *t* divided by the machine system operating time *T*, $label_{HI,t}=t/T$. Assuming that G datasets are required in the training process, the loss function $L^{g}{}_{HI}$ from the g-th training dataset is the mean squared error of $e_{HI,t}$ and $label_{HI,t}$. The naive average induces label imbalance because the length of the dataset varies. An adaptive weighting scheme [32] is introduced to avoid the label imbalance problem, and the formula is
(7)$$L_{HI}=\sum_{g=1}^{G}\exp(L^{g}{}_{HI})L^{g}{}_{HI}/\sum_{g=1}^{G}\exp(L^{g}{}_{HI}).$$

**Domain discriminator.** The domain discriminator consisted of an MLP with one hidden layer connected to the learnable patch of the encoder output. The number of domain discriminators is equal to the number of degradation-process datasets. The output of each domain discriminator was a 2D vector. The second and first elements represent the current inputs sampled during the degradation process. The network learns a domain-invariant HI representation if the domain discriminator cannot differentiate the current input from the dataset.

Assuming that this network has G domain discriminators, the loss function $L^{g}{}_{D}$ of a single-domain discriminator g is based on cross-entropy loss. The same adaptive weighting scheme was applied to make domain discriminators available. A gradient reversal layer is inserted between the domain discriminator and the learnable patch of the encoder output. In the forward process, the gradient reversal layer performs nothing; however, in the backward process, the gradient is multiplied by a pre-specified negative constant −λ. The pre-specified negative constant −λ is followed by −λ=−(2/(1+exp(−10training_process))−1) in training, where training_process denotes the training progress linearly changing from zero to one.

Table 1 shows the network structure parameter setting of TSTN.

### 3.2. Data Pre-Processing

For the data pre-processing part, there are two sub-parts: signal collection and the decomposition of patches. Figure 5 displays the data pre-processing input network.

**Signal collection.** The input of the proposed TSTN was a clip Xt∈ℝm×512 consisting of m frames with 512 spectrum features extracted from the measured vibration signal. The frames were divided according to the time to obtain abundant temporal information. The time-divided relationship follows tindex=τ×[(sin(m−1−indexm−1×π−π2)+1)/2], index=(0, 1, 2, ⋯, m−2, m−1), which τ denotes the time required to collect data.**Decomposition of patches.** Each spectrum feature is decomposed into non-overlapping patches with a size of p; that is, n=512/p. These patches are then flattened into a vector X∈ℝ(m×n)×p as the network input.

In summary, the data preprocessing process can be divided into the following seven steps:(1)Index collection: Assuming that the total length of the time series is 20 s, set parameter m = 5. The indexes for collecting data are 0, 5, 10, 15, and 20;(2)Calculation of times: From the indexes, we can calculate the $t_{index}$ using the index.(3)Sampling data: Based on the calculated $t_{index}$, the data are sampled at these times;(4)Fourier transform: Perform Fourier transform on the sampled time;(5)Select data points: From the Fourier transformed data, select the first 512 points for each sampling time;(6)Divide into blocks: Divide the selected 512 data points into 4 blocks, each with a length of 1278;(7)Reverse concatenation: concatenate these 4 blocks in reverse order.


### 3.3. TSTN Training

This section mainly introduced the proposed diagnosis framework. First, the problem description is illustrated. The proposed machine monitoring methodology is based on historical data, fitting the normalized RUL label $y_{i}$ (1-0) via the input features $x_{i}$. Then, the transfer task is utilized to extract the domain invariant part for cross-operation condition monitoring. The prognostics process consists of two steps: first, constructing the TSTN network based on the input spectrum feature combined with the health indicator; second, using the Monte Carlo method to predict RUL via the TSTN output HI. In this section, a TSTN is developed to predict the machine HI. Details of the proposed TSTN network are presented and shown in Figure 6. The domain discriminator of the developed TSTN was utilized only in the TL training process.

In actual applications, the output of the HI estimator is the machine-condition monitoring HI of the proposed framework. This study utilized the Monte Carlo method based on a linear model with exponential smoothing with parameter 0.9 to generate the downstream prognostics result.

## 4. Experiment Details

### 4.1. Training and Testing Regimes

**Training regime.** Stochastic gradient descent (SGD) with 0.9 momenta is the optimizer in this work. For practical training, the learning rate throughout the training varied according to the following equation:(8)$$\mu =\min\left(S^{-0.5},\;S\times W_{S}{}^{-1.5}\right)/{\left(1+10T_{p}\right)}^{0.75},$$
where S is the number of current training steps, and $W_{S}=1000$. $T_{p}$ is a training process that linearly changes from 0 to 1. The batch size is set to 32, the network weights are updated with gradient accumulation during training, and the random seed is 66. 

**Testing regime.** Once the network finishes training, the testing data are fed into the grid for testing. Apart from performing data pre-processing, other operations are not required for testing. The HI estimator output was the bearing health condition of the input data. The HI-estimated output of the proposed method is $e_{HI,t}$.

### 4.2. Prognostics Result

The validation dataset was obtained from the PRONOSTIA [33] experimentation platform to test and validate bearing fault detection, diagnostic, and prognostic approaches. The rig bench is presented in Figure 7. When the test rig was initialized, a file that contained a 0.1 s vibration signal with a sampling frequency of 25.6 kHz was generated and recorded every 10 s. Three operating conditions were considered; each had two training sets and several testing sets. Information on the training and testing sets is presented in Table 2. The dataset provides 6 sets of data that ran to failure for the establishment of the prediction model, which are 1-1, 1-2, 2-1, 2-2, 3-1, and 3-2. In addition, 11 datasets are provided for RUL, which are 1-3, 1-4, 1-5, 1-6, 1-7, 2-3, 2-4, 2-5, 2-6, 2-7, and 3-3.

The scoring benchmark was set according to [30], and only the vertical vibration signal (2560 points per file) was used to generate the network output. The size of the spectrum generated via fast Fourier transform was 512. The pre-processing operation entailed 21 spectrum frames, and each structure was decomposed into eight non-overlapping patches. The training epoch was set to 60. To achieve cross-domain condition monitoring in the bearing, we use six training datasets in the same training process.

After finishing the training process of the proposed network, the network can be utilized to monitor the health condition of the bearing in the testing data. The proposed method’s expected output $e_{HI,t}$ is a direct HI of 0 to 1. To demonstrate the capability of the direct HI in RUL prediction, we use the Monte Carlo method based on the linear model for curve fitting and RUL prediction ($pre_{RUL,t}$).

Figure 8 shows the estimated HI results and RUL predictions from the test data of the proposed method. The blue line represents the HI output of the proposed method. The green line refers to the RUL prediction and 95% confidence interval, and the yellow area represents the probability distribution function of the RUL prediction result $pre_{RUL,t}$. As shown in Figure 8, HI estimation using the proposed method can effectively capture the bearing degradation trend. The proposed method can provide nearly linear HI estimation.

## 5. Comparisons and Analysis

Then, the normalized prediction error $Er_{i}$ and benchmark scores were calculated [33]. The results of all the testing sets are listed in Table 3. The specific calculation formula is as follows:(9)$$\%Er_{i}=100\times \frac{ActRUL_{i}-\hat{RUL_{i}}}{ActRUL_{i}}$$
(10)$$A_{i}=\{\begin{array}{c} \begin{array}{cc} \exp^{-\ln(0.5)\cdot (Er_{i}/5)} & if\;Er_{i}\;\leq \;0 \end{array}\\ \begin{array}{cc} \exp^{+\ln(0.5)\cdot (Er_{i}/20)} & if\;Er_{i}\;>\;0 \end{array} \end{array}$$
(11)$$Score=\frac{1}{11}\sum_{i=1}^{11}\left(A_{i}\right)$$

As presented in Table 3, except for testing sets 2-7 and 3-3, the RUL prediction results of the proposed method are reasonable. The errors in the prediction results of datasets 1-5 to 2-6 were shallow, and the proposed method could effectively perform bearing condition monitoring with testing sets 1-5, 1-7, 2-4, and 2-6. Compared to the RNN-based RUL prediction method [34], convolutional LSTM network [35], Bi-directional LSTM network with attention mechanism [36], and the traditional RUL prediction method based on vibration frequency anomaly detection and survival time ratio [37], the proposed TSTN method has higher RUL prediction accuracy. These results confirm that the proposed method is applicable to the prognostics of mechanical rotating components. For the last two datasets, the RUL predictions exhibit large deviations. The reason for these large deviations is that the vibration signal changes slightly only in the early degradation process, which displays a linear degradation trend. However, as time goes on, the linear trend becomes nonlinear. The HI $e_{HI,t}$ does not have a linear change rate in the latter stage. Hence, the proposed HI is unsuitable for predicting the RUL in latter-stage degradation. However, compared with other methods, the computational complexity is higher, and the training time is 3 h.

**Table 3 sensors-23-09190-t003:** RUL Prediction results of the proposed method.

Dataset	$Er_{i}\%$ (Our)	$Er_{i}\%$ [34]	$Er_{i}\%$ [37]	$Er_{i}\%$ [36]	$Er_{i}\%$ [35]
1-3	0.5	43	37	−5	55
1-4	23	67	80	−9	39
1-5	25	−22	9	22	−99
1-6	9	21	−5	18	−121
1-7	−2	17	−2	43	71
2-3	82	37	64	45	76
2-4	85	−19	10	33	20
2-5	2	54	−440	50	8
2-6	70	−13	49	26	18
2-7	−1122	−55	−317	−41	2
3-3	−1633	3	90	20	3
Score	0.4017	0.2631	0.3066	0.3198	0.3828

### Discussions of the Proposed Methodology

**Influence of multi-head number.** To improve the learning capability of the self-attention layer of the encoder, linearly project keys, values, and query h times, which is called the multi-head attention operation. In this section, the influence of multi-head numbers is discussed. The predicted RUL benchmark scores of different multi-head numbers indicate that 16 (score is 0.4017) is the most suitable for the prognostics task, and it is higher than the results of four multi-head (score is 0.0607) and eight multi-head (score is 0.1124) numbers. Theoretically, the larger the multi-head number, the stronger the fitting capability. However, the rotary position embedding method requires almost four numbers to indicate location information. When the multi-head operation breaks up the rotary position embedding, the self-attention calculation cannot capture the time information. Therefore, the score of the 32 multi-head numbers was 0.2631, and that of the 64 multi-head numbers was 0.0689. In summary, the multi-head number needs to be set to $d_{\mathrm{model}}/4$ in the prognostics task.

**Discussions with/without transfer learning.** The proposed method uses the domain discriminator with the gradient reversal layer to extract the domain-invariant RUL representation. We expect to use the TL method to improve the linearity of the estimated HI under different operating conditions. An experiment was conducted on a TSTN without a TL, reflecting the domain discriminator’s effectiveness in cross-operating condition monitoring. Aside from removing the domain discriminator, the other network framework settings were similar to those in Figure 9. The RUL prediction score decreased from 0.4017 to 0.0515. The prognostic results of TSTN and TSTN without a domain discriminator for test datasets 1-6, 1-7, 2-4, and 2-6 indicate TL’s effectiveness. Figure 9 shows the comparison of TSTN and TSTN without transfer learning. The blue lines represent the classical TSTN HI results, and the greenish-blue lines denote the HI-estimated effects of TSTN without TL. TL improves the TSTN prognostics capability in cross-operating condition situations.

**Effectiveness of the self-attention mechanism.** This study utilized test sets 1-6 to generate a self-attention heatmap (shown in Figure 10) to indicate the effectiveness of the self-attention mechanism. The longitudinal of the self-attention heatmap refers to the m time frames, and the transverse of the self-attention heatmap pertains to the 16 multi-heads with eight patches. In this study, 1/3, 2/3, and 1 of the normalized operating time were selected. When a patch has a high self-attention value, the network focuses on that patch. Figure 10 shows that only a few heads undertake the HI estimation task, but our previous study indicated that a sizeable multi-head number equates to strong learning capability. A possible reason is that a large multi-head results in a flexible feature association capability, which means that features can be selected precisely.

The first self-attention layer was a long-term self-attention layer. In Figure 10, head 12 of long-term self-attention captures the severe degradation at the end of the operating time, and head 4 focuses on the weak degradation at the early and middle operating stages. After the long-term self-attention layer, the spectrum long-term change relationship was obtained, and the local self-attention layer was used to capture abundant information in one frame. In Figure 10, a clear degradation relationship was captured. Head 11 of the local self-attention layer captured the weak degradation in the early operating stage. Head 10 focuses on degradation in the middle operating phase, and head 13 focuses on rapid degradation at the late operational stage. Figure 10 shows that local self-attention plays a greater role than the long-term self-attention layer. However, the learning capability sharply declined when the two layers’ order was changed. This result indicates that the long-term self-attention layer generates the long-term relationship and is strengthened by the local self-attention layer. 

In summary, the multi-heads in the short-term self-attention layer focus on the spectrum value, thereby making the proposed TSTN sensitive to spectrum value changes.

## 6. Conclusions

Machine prognostics play a crucial role in the automaticity and intelligence of industrial plants, especially in intelligent plant manufacturing and asset health management. This study proposed a TSTN-based machine prognostic method to solve the HI automatic construction with a high-dimensional feature input in a cross-operating condition. The proposed method is integrated with a novel transformer network structure with a domain adversarial TL consisting of an encoder, an HI estimator, and a domain discriminator. First, the proposed TSTN automatically extracts features (HI) from a long-term high-dimensional feature input, avoiding information loss caused by manual feature extraction. Second, we have devised a self-attention mechanism that encompasses long-term, short-term, and local perspectives, enabling it to discern the dynamic interplay between long-term and short-term machine health conditions. Third, when incorporating the DAN TL method, it addresses issues of cross-operating conditions and different data distributions. The domain discriminator with a gradient reversal layer can generate an accurate and robust HI. Compared to the RUL prediction methods based on RNN, the convolutional LSTM network, the bi-directional LSTM network with an attention mechanism, and traditional strategies rooted in vibration frequency anomaly detection and survival time ratios, our proposed TSTN approach achieves a superior score of 0.417, indicating its enhanced accuracy in RUL prediction. In the future, we plan to collect more datasets to verify the effectiveness of the proposed method. In addition, we will conduct further research on improving the generalization ability of the method for dealing with extremely cross-operating conditions, such as predicting the RUL for an unseen operating condition. The proposed method is a promising methodology for coping with HI estimator construction with a high-dimensional feature input, monitoring machine health conditions, and predicting machines’ RUL in cross-operating working conditions.

## Figures and Tables

**Figure 1 sensors-23-09190-f001:**
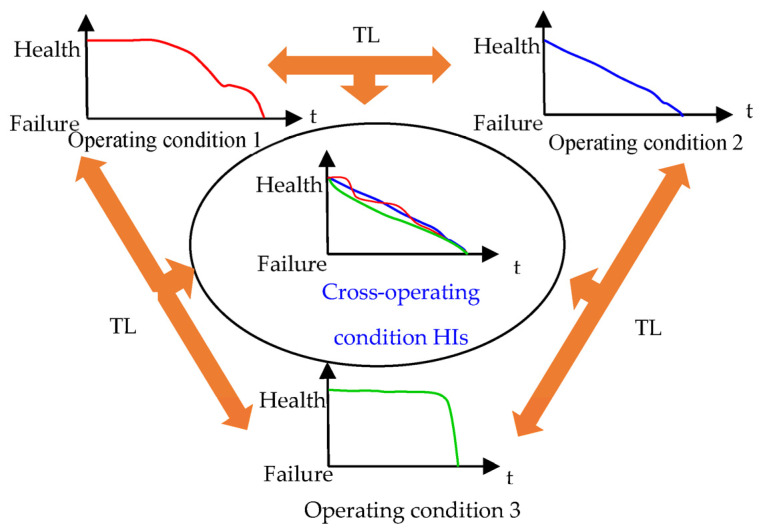
Transfer learning for cross-operating condition HIs construction.

**Figure 2 sensors-23-09190-f002:**
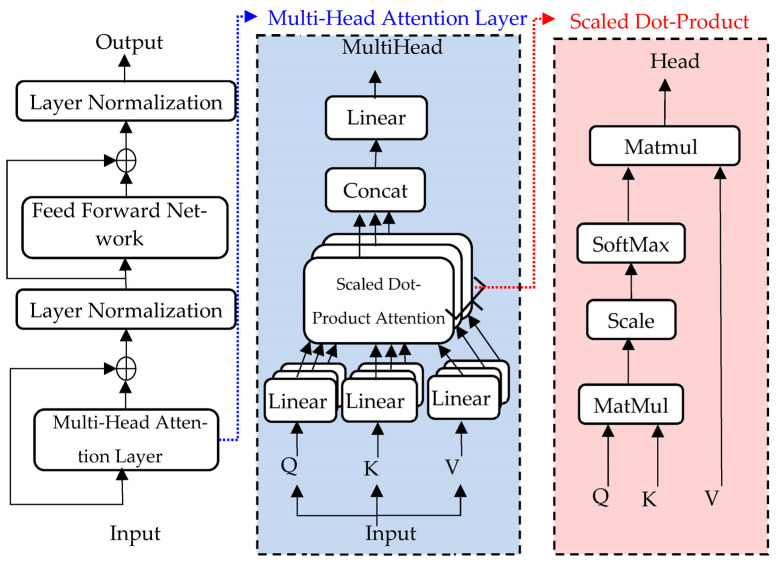
Details of the self-attention layer network structure.

**Figure 3 sensors-23-09190-f003:**
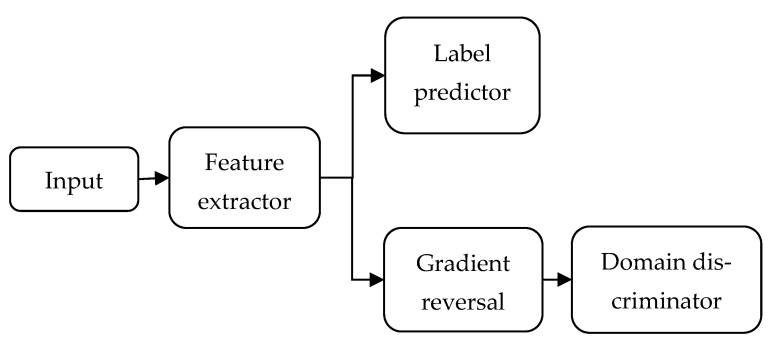
The architecture of an adversarial domain network.

**Figure 4 sensors-23-09190-f004:**
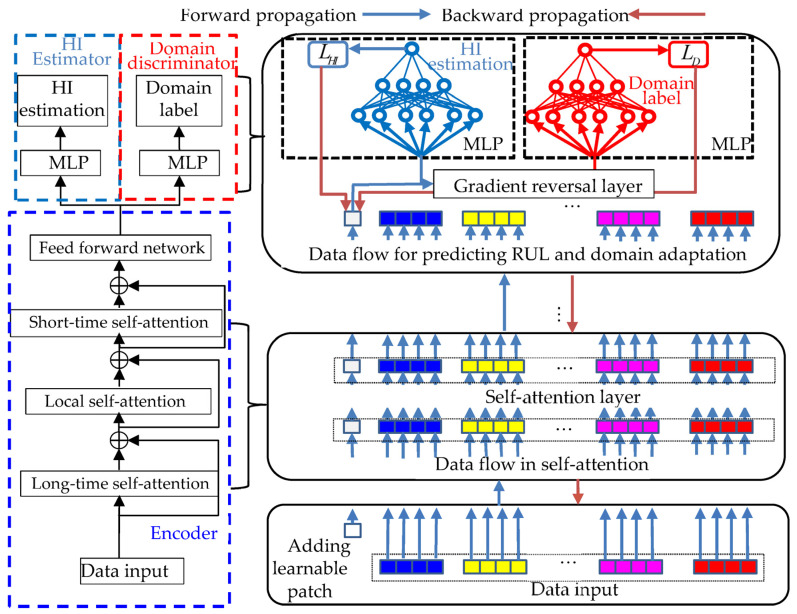
The whole flowchart of the proposed TSTN methodology.

**Figure 5 sensors-23-09190-f005:**
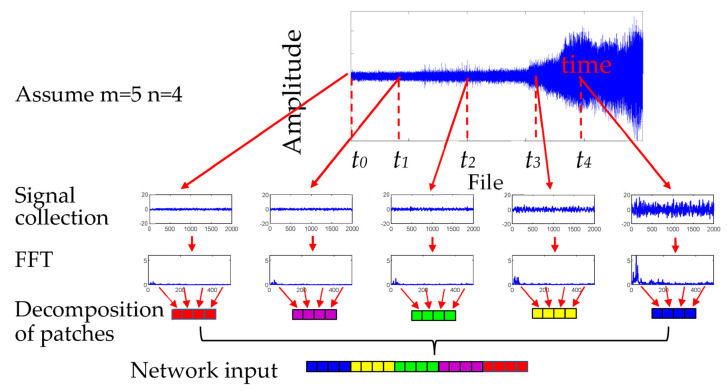
Data pre-processing input network.

**Figure 6 sensors-23-09190-f006:**
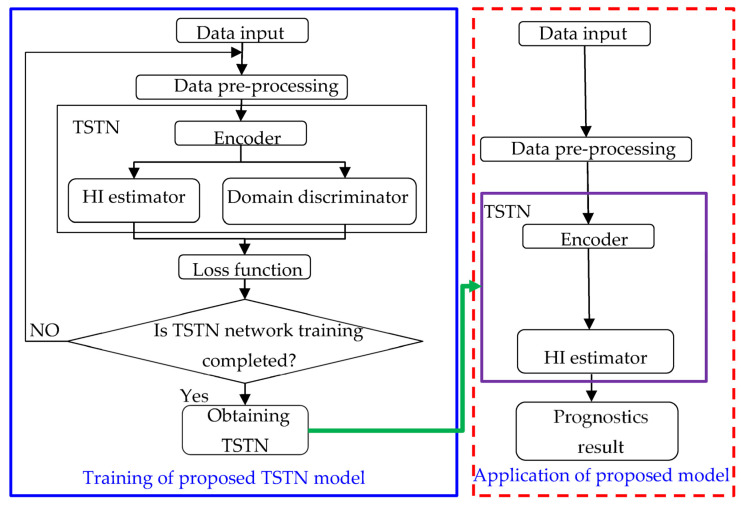
Flowchart of the proposed TSTN model and its application.

**Figure 7 sensors-23-09190-f007:**
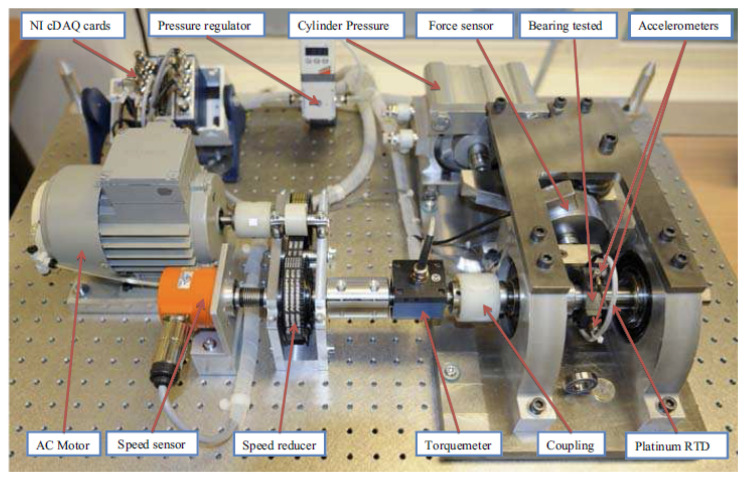
Overview of PRONOSTIA.

**Figure 8 sensors-23-09190-f008:**
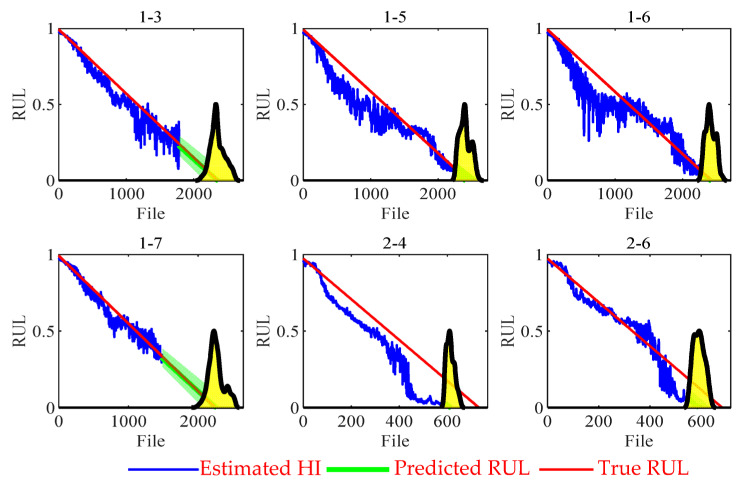
Estimated HIs of the proposed method. (Yellow areas represents the RUL of probability density distribution).

**Figure 9 sensors-23-09190-f009:**
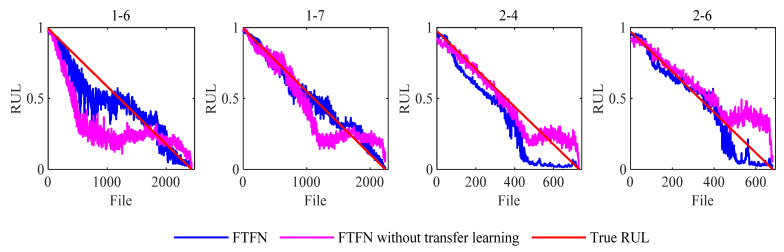
Comparison of TSTN and TSTN without transfer learning.

**Figure 10 sensors-23-09190-f010:**
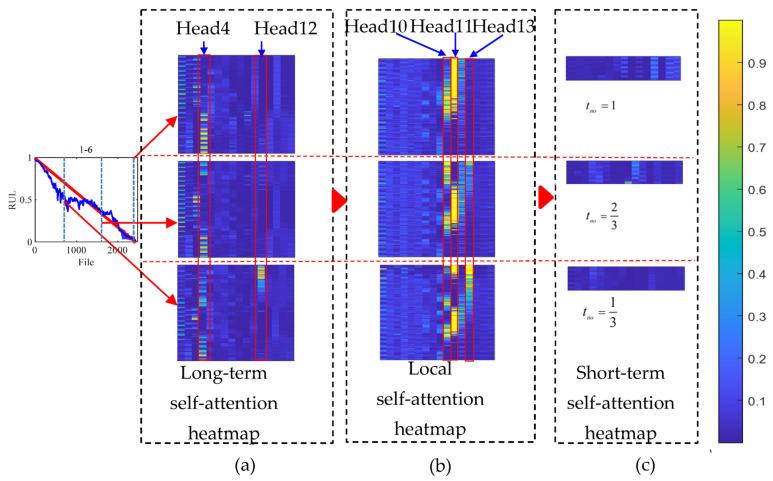
The long-term, local, and short-term self-attention heatmap from testing sets 1-6. (**a**) Long-term self-attention heatmap; (**b**) Local self-attention heatmap; (**c**) Short-term self-attention heatmap.

**Table 1 sensors-23-09190-t001:** Parameter setting of TSTN.

Encoder	Multi-Head	$d_{\mathrm{model}}$	$d_{\mathrm{diff}}$	Dropout rate
	16	64	512	0.2
HI estimator (MLP)	Layer	Dense	Activation function	number
	Fully connected	32	GeGLU	1
	Fully connected	1	GeGLU	1
Domain discriminator (MLP)	Layer	Dense	Activation function	number
	Fully connected	32	GeGLU	Equal to dataset number
	Fully connected	2	Softmax	

**Table 2 sensors-23-09190-t002:** Information on the FEMTO-ST dataset.

Operating Condition	1	2	3
Speed (rpm)	1800	1650	1500
Loading (N)	4000	4200	5000
Training dataset	1-1, 1-2	2-1, 2-2	3-1, 3-2
Testing dataset	1-3, 1-4, 1-5, 1-6, 1-7	2-3, 2-4, 2-5, 2-6, 2-7	3-3

## Data Availability

The data presented in this study are openly available in Ref. [33].

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
