# Peer review of "Machinery Prognostics and High-Dimensional Data Feature Extraction Based on a Transformer Self-Attention Transfer Network"

_sensors, 2023, doi:10.3390/s23229190_

Round 1
Reviewer 1 Report (New Reviewer)
Comments and Suggestions for Authors
Instead of calling it experimental analysis, it can be Experimental Details
Give some background details as to how the data was grouped as 1-1, 1-2, 1-3 etc.?
In page 11 of 17, it is not "in Figure 7", but Table 2!
In fig. 8, the last footer is not clear, "Ture RUL"?
In Table 3, how was the SCORE calculated, how do you interpret this score?
Highlight the significance of your work in terms of practical application in Machinery prognostics, the results presented and discussions presented are more theoretical / research oriented rather than practical / outcome oriented. In some words explain how use of the proposed method helps in doing Machinery prognostics by determining the RUL.
Author Response
Thank you very much for your comment. I have made a point-to-point response. Please see the attachment document.

Reviewer 2 Report (New Reviewer)
Comments and Suggestions for Authors
I found your article very interesting titled “Machinery Prognostics and High-Dimensional Data Feature Extraction Based on a Transformer Self-Attention Transfer Network”, but in my opinion below remarks would improve your manuscript under the scientific level.
Comments and Suggestions for Authors:
1. In the Abstract, please refer to the main outcomes of the experiment. Only the methods and general knowledge are mentioned there.
2. In the manuscript I’ve spotted several different types of fonts. Please unify it.
3. In the Introduction I miss the applications regarding using the Machine Learning in the prognosis of degradation. I suggest to add following papers to enrich your list of references:
· Machine Learning Techniques for Multi-Fault Analysis and Detection on a Rotating Test Rig Using Vibration Signal, Symmetry 2023, 15(1), 86
· Intelligent Diagnostics of Radial Internal Clearance in Ball Bearings with Machine Learning Methods, Sensors 2023, 23(13), 5875.
· Improved Fault Classification for Predictive Maintenance in Industrial IoT Based on AutoML: A Case Study of Ball-Bearing Faults, Processes 2023, 11(5), 1507.
4. Figure 5 requires quality improvement.
5. Figure 7, there is misspelling in the caption.
6. Figure 8, how can you explain the probability distribution?
7. The conclusions must be finished with the quantified description of the results. In the current form, there is only plain description.
Comments on the Quality of English Language
Only minor changes are requires in the text.
Author Response
Thank you very much for your comment. I have made a point-to-point response. Please see the attachment document.

Round 2
Reviewer 2 Report (New Reviewer)
Comments and Suggestions for Authors
All proposed remarks have Beenhakker introduced into the revised version of the mamuscipt. I reccomend the paper for its publishing in the present form.
Yours faithfully
Reviewer
This manuscript is a resubmission of an earlier submission. The following is a list of the peer review reports and author responses from that submission.
Round 1
Reviewer 1 Report
Comments and Suggestions for Authors
Main comment:
This paper presents a transformer-based self-attention transfer learning network (TSTN) for machinery prognostics. The TSTN utilizes long-term, local, and short-term self-attention mechanisms to capture crucial time-varying information from high-dimensional vibration signals. It also incorporates a domain discriminator to extract invariant features from different machine operating conditions. The TSTN has been shown to process high-dimensional vibrational datasets, monitor cross-operating conditions, and accurately predict remaining useful life (RUL) when combined with the Monte Carlo method.
The article proposes a relatively new method that has been validated and has a certain degree of feasibility and novelty. However, some issues need to be comprehensively corrected before the article can be published
comments:
1. Provide a more detailed explanation of the self-attention transfer learning network (TSTN) and its underlying principles. This will help readers understand the theoretical foundation of the proposed method.
2. Discuss the limitations and assumptions of the TSTN approach. This will provide a more comprehensive understanding of the method and its applicability to different scenarios.
3. Include a detailed description of the datasets used in the experiments, including their sources, characteristics, and any preprocessing steps applied. This will enhance the reproducibility of the results.
4. Conduct a comparative analysis with existing prognostics methods to demonstrate the superiority of the proposed TSTN approach. This will provide a benchmark for evaluating the performance of the method.
5. Perform sensitivity analysis on the hyperparameters of the TSTN, such as the number of attention heads, the learning rate, and the batch size. This will help determine the optimal configuration for achieving the best performance.
6. Improve the clarity and organization of the paper by providing clear section headings and subheadings. This will make it easier for readers to navigate through the content.
7. Discuss the potential applications and practical implications of the proposed TSTN approach. This will help readers understand the real-world relevance of the research.
8. Address the limitations and challenges of implementing the TSTN approach in practical scenarios, such as computational requirements and data availability. This will provide a more realistic perspective on the feasibility of the method.
9. Provide recommendations for future research directions, such as exploring different types of machinery datasets, investigating alternative feature extraction techniques, and evaluating the scalability of the TSTN approach. This will inspire further advancements in the field.
Comments on the Quality of English Language
Moderate editing of English language required.
Reviewer 2 Report
Comments and Suggestions for Authors
General Remarks:
In this research, the authors have applied the Machinery degradation assessment can offer meaningful information for prognosis and 11 health management. The health indicator (HI) construction is crucial in assessing the machinery's health degradation. However, building a useful health indicator directly from high-dimensional raw vibration signals is still challenging, especially under cross-operating conditions. The current HIs with high-dimensional dataset input are inefficient in determining the degradation trend and providing an accurate remaining useful life (RUL) prediction. This paper presents a transformer-based self-attention transfer learning network (TSTN) to construct an HI consisting of an encoder, an estimator, and a domain discriminator. First, the encoder is designed with long-term, local, and short-term self-attention mechanisms to capture crucial time-varying information from a high-dimensional dataset. Second, the estimator can map the embedding from the encoder output to the estimated degradation trends. Third, the domain discriminator is proposed to extract invariant features from different machine operating conditions. Case studies have demonstrated that the proposed TSTN can process a high-dimensional vibrational dataset, monitor cross-operating conditions, and predict accurate RUL when combined with the Monte Carlo method.
The following points need to be addressed
1. The conclusion must answer whether the proposed method can solve the research problem and achieve the objective. How can the numerical approach answer the existing issues? What is the most important result? What are the implications for science and technology development? What is the novelty? What is the objective of this research? It is missing in the whole draft. Add more detail about figure 2. Table 1 should be explain
2. What software is used for the simulations? Was the code for the implemented by the authors or a function already existing in the software was used? If the code for the numerical method was taken from another publication or is part of the software used, please cite the resource.
3. The Introduction should make a compelling case for why the study is useful along with a clear statement of its novelty or originality by providing relevant information and providing answers to basic questions such as: What is already known in the open literature? What is missing (i.e., research gaps)? What needs to be done, why, and how? Clear statements of the novelty of the work should also appear briefly in the Abstract and Conclusions sections
4. The last paragraph of the introduction states a few research questions but the conclusion is not at all reflect those. It is suggested that the conclusion is modified so that it connects to the research question posed in the introduction section.
Comments on the Quality of English Language
Moderate editing of English language required
Reviewer 3 Report
Comments and Suggestions for Authors
This paper introduces a health indicator predictor of machinery based on transfer learning and self-attention mechanism. The originality of this work is not enough. In addition, this paper is not well-organized. The detailed comments are as follows:
1. The technical contribution of this work is not sufficient. It seems that the proposed method is just a fusion of some existing algorithms.
2. Some key details are not presented clearly, especially descriptions of data pre-processing and query-key-value computation.
3. The authors should check the normality of mathematical notation in the manuscript.
4. The content in Fig. 9 is overlapped.
Comments on the Quality of English Language
The English writing of this paper is terrible.
Round 2
Reviewer 1 Report
Comments and Suggestions for Authors
All my concerns have been addressed.
Reviewer 3 Report
Comments and Suggestions for Authors
After reading the revised manuscript, I have to stand by my previous opinion. This is because the work in the paper is not sufficiently innovative from a methodological point of view. I don't think combining Transformer and DAN is a novel original work.
Comments on the Quality of English Language
English writing of this manuscript needs to be improved